# Elucidation of the viral disassembly switch of tobacco mosaic virus

Felix Weis[1,†] (iD), Maximilian Beckers[1,2,†] (iD), Iris von der Hocht[3,4] (iD) & Carsten Sachse[1,3,4,*] (iD)

## Abstract

Stable capsid structures of viruses protect viral RNA while they also require controlled disassembly for releasing the viral genome in the host cell. A detailed understanding of viral disassembly processes and the involved structural switches is still lacking. This process has been extensively studied using tobacco mosaic virus (TMV), and carboxylate interactions are assumed to play a critical part in this process. Here, we present two cryo-EM structures of the helical TMV assembly at 2.0 and 1.9 Å resolution in conditions of high $Ca^{2+}$ concentration at low pH and in water. Based on our atomic models, we identify the conformational details of the disassembly switch mechanism: In high $Ca^{2+}$/acidic pH environment, the virion is stabilized between neighboring subunits through carboxyl groups E95 and E97 in close proximity to a $Ca^{2+}$ binding site that is shared between two subunits. Upon increase in pH and lower $Ca^{2+}$ levels, mutual repulsion of the E95/E97 pair and $Ca^{2+}$ removal destabilize the network of interactions between adjacent subunits at lower radius and release the switch for viral disassembly.

**Keywords** Caspar carboxylates; cryo-EM; helical reconstruction; tobacco mosaic virus; virus assembly/disassembly

**Subject Categories** Microbiology, Virology & Host Pathogen Interaction; Structural Biology

## Introduction

The capsid of RNA viruses provides a container that ensures the protection of the viral genome from degradation in the extracellular environment. The shell of tobacco mosaic virus is particularly thermostable in ambient temperatures and resistant to degradation across a wide range around neutral pH [1]. The TMV capsid is primarily composed of a 17 kDa coat protein (CP) that is organized in a helical assembly thereby tightly enclosing the viral RNA of the genome [2–5]. TMV enters the plant cell through mechanical lesions that transiently open the outer membrane [6]. The propagation of the virus is then triggered by a controlled opening of the viral capsid inside the host cell in order to facilitate cotranslational virion disassembly by the replisome [7]. This requires a molecular mechanism that is capable of sensing the differences between the extracellular and the intracellular medium, followed by destabilization of the assembled virion and subsequent replication in the host cell. A series of plant viruses use $Ca^{2+}$ and pH triggered disassembly [8] and thereby exploit the lower $Ca^{2+}$ and proton concentrations inside the plant cell compared with the extracellular environment [8].

The disassembly behavior of TMV has been biochemically studied in remarkable detail. Based on titration experiments, it was reported that TMV contains groups that titrate with $pK_a$ values between 7 and 8 [9], leading to the hypothesis that a carboxylate cluster binds protons with high affinity. These putative residues termed "Caspar-Carboxylates" [9,10] are thought to drive virus disassembly through mutual repulsion upon entering the intracellular environment. In addition, it has also been demonstrated that TMV binds $Ca^{2+}$ via these residues [11,12]. The CP folds into a four-helix bundle that is commonly divided into three regions referring to the radial distance from the helical axis [13]: lower radius 20–40 Å, middle radius 40–60 Å, and higher radius 60–90 Å. Mutational studies have identified the critical residues [14,15] involved in the disassembly process: E50 and D77 at middle radius of the cylinder cross section have been hypothesized to be involved in axial carboxylate interactions, whereas E95, E97, and E106 at lower radius mediate lateral carboxylate and possible $Ca^{2+}$ interactions [4]. Moreover, it has been shown by single molecule force spectroscopy, that upon $Ca^{2+}$ decrease the 5′ end of the capsid becomes exposed and RNA-coat protein interactions are weakened at the rest of the virion [7], presumably to advance cotranslational disassembly by the replisome machinery.

Despite the wealth of biochemical and mutational studies, our structural understanding of the conformational switch sensing the environmental changes is still incomplete. Although TMV was subject to a plethora of structural studies [13,16–18], resolution of the helical rod was limited to 2.9 Å when determined by early X-ray fiber diffraction [5] or later to 3.3 Å by electron cryo-microscopy

1 Structural and Computational Biology Unit, European Molecular Biology Laboratory (EMBL), Heidelberg, Germany
2 Faculty of Biosciences, EMBL and Heidelberg University, Heidelberg, Germany
3 Ernst-Ruska Centre for Microscopy and Spectroscopy with Electrons 3/Structural Biology, Forschungszentrum Jülich, Jülich, Germany
4 JuStruct: Jülich Center for Structural Biology, Forschungszentrum Jülich, Jülich, Germany
*Corresponding author. E-mail: c.sachse@fz-juelich.de
†These authors contributed equally to this work

(cryo-EM) studies [13,16] and more recently up to 2.3 Å resolution [19]. At these resolutions, the maps were sufficiently clear for the assignment of the architecture of the CP and annotation of bulky side chains, whereas further details regarding the conformation of the implicated side chains remained undetermined. The proposed Caspar carboxylate residues E95, E97, E106, and the calcium binding site are found in a more flexible part of the protein at lower radius with high B-factors, as it is expected for such a metastable switch. In the absence of RNA within the disk assembly of TMV at higher resolution, the respective residues were not detectable and assumed to be disordered [20]. Generally, negatively charged amino acid residues suffer from faster radiation damage when imaged by cryo-EM [16,21], which makes them more difficult to model. Therefore, the precise structural details of this intricate viral disassembly switch remain to be elucidated.

In order to address this outstanding fundamental question regarding viral RNA disassembly, we used cryo-EM including latest developments of high-resolution imaging, data processing, and map interpretation methods and determined two ∼ 2 Å resolution TMV structures. The maps reveal that the metastable switch is based on a $Ca^{2+}$-sensitive network of carboxylate and iminocarboxylate residues at lower radius, which become destabilized by $Ca^{2+}$ release at higher pHs. The cryo-EM structures captured TMV at different conformational states of this network and thereby directly reveal the transitional mechanics of the switch driving viral disassembly.

## Results and Discussion

### Cryo-EM structures of TMV at 1.9 and 2.0 Å resolution

To elucidate different structural states of the viral assembly, we prepared TMV in two different conditions, first in the presence of 20 mM $CaCl_2$ at pH 5.2 (referred to as $Ca^{2+}$/acidic pH) and second in water in the absence of any cations. Both samples were plunge-frozen and imaged using a 300 kV electron microscope equipped with a GIF Quantum K2 camera (Fig 1A). With the collected micrographs, we determined the 2.0 and 1.9 Å resolution TMV structures in $Ca^{2+}$/acidic pH and water, respectively (Fig EV1A). Both maps show local resolutions up to 1.8 Å for the CP core and ∼ 5 Å for the disordered C-terminal tail (Figs 1B and EV2). Map details agree with the expected high-resolution features such as defined carbonyl oxygens of the protein backbone. The map can also be used to locate non-protein components such as water molecules and metal ions. In order to minimize the influence of noise during molecule and ion placements, we used recently developed confidence maps at a 1% false discovery rate (FDR) threshold [22] that is known to suppress noise in comparison with fixed sigma thresholds of EM maps. Using these confidence maps together with expected donor–acceptor hydrogen bond lengths [23] (Fig EV2D), we modeled a total of 92 water molecules for TMV in water and 71 water molecules under $Ca^{2+}$/acidic pH. Due to the proximity to RNA, we modeled 4 $Mg^{2+}$ ions bound to RNA as well as well-defined side-chain conformers per CP in both conditions (Fig 1C–E). A critical $Ca^{2+}$ could be located in the $Ca^{2+}$/acidic pH structure whereas no $Ca^{2+}$ binding was observed at the proposed $Ca^{2+}$ site at the RNA [5,24] in both maps (Fig EV3A). In order to verify the principal biological activity of the imaged virus structures, we demonstrated infectivity of the used virus batch in tobacco plants, which showed typical symptoms like stunted growth and necrotic lesions 35 days post-infection (Fig EV4).

### $Ca^{2+}$/acidic pH and water structures at lower radius

Using our recently developed statistical framework for the annotation of molecular map features [22], i.e., confidence maps that assist in the assignment of atomic models within weaker cryo-EM map density, we were able to analyze the CP map from $Ca^{2+}$/acidic pH and water samples in a comparative manner (Fig 2A). Although the two determined TMV maps are very similar for most of the CP, the lower radius region differs significantly at an FDR of 1% (Fig 2A right). Comparison of this lower radius map features with recently determined EM map features EMD2842 [16] (Fig EV2B) revealed that previous studies only poorly resolved this part of the protein. Detailed comparison of the $Ca^{2+}$/acidic pH and water structure at the lower radius region showed that the protein backbone follows a different path (Fig 2B top). In the determined $Ca^{2+}$/acidic pH structure, we built an atomic model matching the map (Fig 2B left). In the water structure at lower radius, however, we identified three co-existing models in the residue range 97–100 that describe the map, e.g., the map of the water structure is consistent with multiple conformations of E97 (Fig 2B right).

In order to analyze the two structures in more detail, we refined the atomic coordinates by a common real-space optimization approach (Table 1) [25]. As expected, the structural differences between the $Ca^{2+}$/acidic pH and water structures in the region outside the 90–110 region are very low with RMSDs below 0.5 Å whereas higher between 2.4 and 2.8 Å inside the lower radius region. The multiple models placed for the water structure in the lower radius region (E97–A100) deviate to a smaller but significant extent around 1.0 Å (Table 2). It should be noted that the cryo-EM density for these three models is not discrete but they are compatible with a flexible ensemble of models describing the continuous density. Closer inspection of the lower radius interface between neighboring subunits reveals additional differences: In the $Ca^{2+}$/acidic pH structure, map features of a bound ion are present and consistent with coordination by E106, N101, N98, and a backbone carbonyl oxygen. In the water structure, however, this complete coordination is missing and asparagines N101 and N98 are facing away from the central map features. In order to confirm the observed differences on the level of the cryo-EM maps, we compared the respective loop regions by difference mapping (Fig EV5A). The difference map between the two conditions shows the rearrangement of the α-helical segment including the presence and coordination of the $Ca^{2+}$ ion (Fig EV5B). Annotation of the $Ca^{2+}$ ion is based on high map values at the respective site and octahedral coordination (Fig EV5C). Moreover, local resolutions plots of the cryo-EM maps justify the placement of side chains and show once more the stabilizing effect of the $Ca^{2+}$/acidic pH condition (Fig EV5D). Therefore, we conclude that under $Ca^{2+}$/acidic pH conditions, this subunit interface is stabilized by a $Ca^{2+}$ ion, whereas in TMV in water the respective site is occupied by water molecules.

### Interactions involved in the metastable switch

Further comparative analysis of the detailed interactions at lower radius revealed an extended helix of the short α-helical segment in

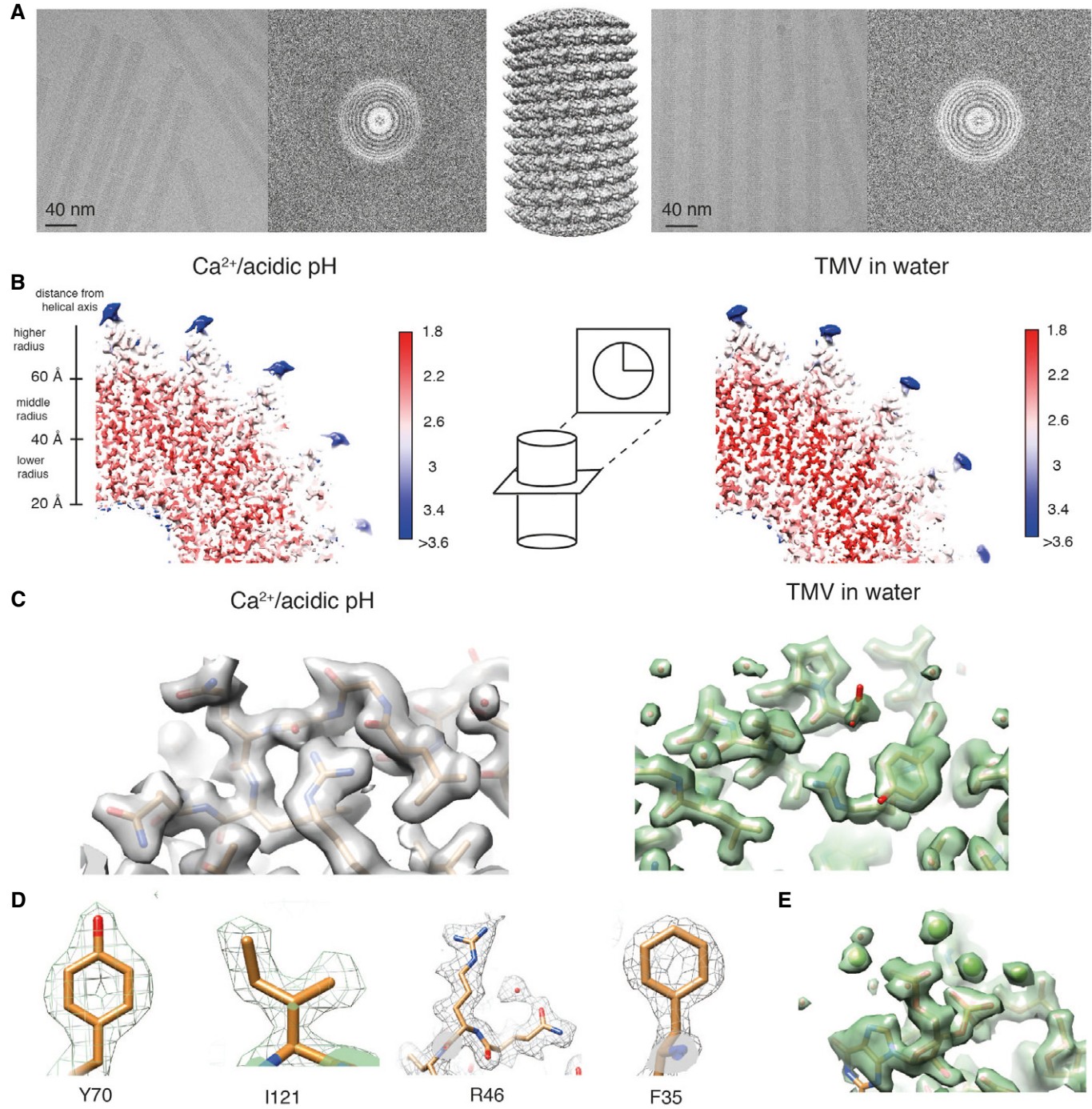

**Figure 1. High-resolution cryo-EM structures of tobacco mosaic virus (TMV) in conditions of $Ca^{2+}$/acidic pH (gray, left) and water (green, right).**

A Characteristic micrographs for both data sets, respectively.

B Local resolutions mapped on the respective 3D reconstructions. The center of the coat protein is resolved up to 1.8 Å with some features of approx. 5 Å resolution at the C terminus.

C Both maps show well-resolved protein features including water molecules.

D Side-chain features of Y70, I121, R46, and F35 residues at $Ca^{2+}$/acidic pH.

E Snapshot of RNA map features with $Mg^{2+}$ ions in water conditions.

the $Ca^{2+}$/acidic pH model by the single residue N98 (Fig 3A). Plots of refined atomic B-factors also show a decrease from 42 to 25 $Å^2$ in the lower radius region for the $Ca^{2+}$/acidic pH condition in comparison with water, supporting the notion that $Ca^{2+}$ stabilizes the assembly structure (Fig EV5E). Next, we more closely examined the carboxylate residues previously identified to be critical in the

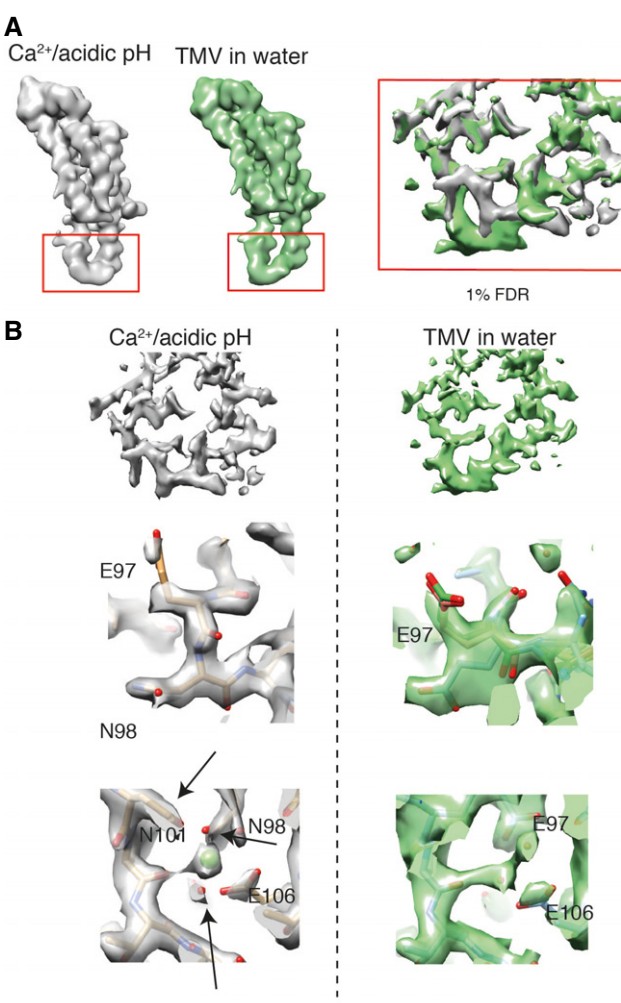

**A** Ca²⁺/acidic pH    TMV in water

1% FDR

**B** Ca²⁺/acidic pH    |    TMV in water

E97

N98

N101    N98

E106

E97

E106

All densities shown at a FDR of 1%

**Figure 2. Confidence maps thresholded at 1% FDR from cryo-EM maps of Ca²⁺/acidic pH and water structure.**

A   Low-pass-filtered monomer map features of Ca²⁺/acidic pH (gray) and water structure (green) with differences at lower radius (left). Zoomed inset (right) with maps displayed at a false discovery rate (FDR) threshold of 1% showing significant differences.

B   Detailed map comparison at lower radius region of Ca²⁺/acidic pH (gray, left column) with the water condition (green, right column). A total of three atomic models (model 1: cyan, model 2: pink, model 3: green) describe the map of TMV in water whereas the Ca²⁺/acidic pH map could be modeled with a single atomic model (center). Located Ca²⁺ ion in the Ca²⁺/acidic pH map with different conformations in the water structure (bottom).

disassembly process (E50, D77, E95, E97, and E106). First at medium radius, E50 and D77 contribute to tight axial carboxylate contacts at a distance of 3.0 Å and showed no differences between the two determined structures. Second at lower radius, we find that glutamates E97 and E95 make up tight inter-subunit interactions in the Ca²⁺/acidic pH structure with a distance between the carboxylate groups of 2.5 Å (Fig 3B left, top). Although E106 is not found in contact with other carboxylates, E106 is involved in the coordination of Ca²⁺. The respective Ca²⁺ site is shared between the adjacent subunits as it is coordinated by E106 and N98 from one CP

monomer and N101 as well as the backbone carbonyl oxygen of P102 from the neighboring monomer (Fig 3B left, bottom). These close inter-subunit interactions between neighboring CPs add to the stability of the helical assembly. The water structure, however, is lacking the close inter-subunit carboxylate contacts as E97 shows two different conformations, one facing toward E106 (model 1) and the other toward E95 (model 2) with significantly longer distances of 3.9 and 5.4 Å between carboxylates, respectively (Fig 3B right, top). Residues N101 and N98 also assume different conformations in the water condition (Fig 3B right, bottom), whereas they participate in the coordination of Ca²⁺ in the Ca²⁺/acidic pH condition. We conclude that due to the loss of Ca²⁺ coordination as well as the buildup of carboxylate repulsion at higher pH, the residue network in proximity of E106, N101, and N98 becomes destabilized and changes conformations (Fig 4).

**Toward a structural mechanism of the viral disassembly switch**

Based on our two high-resolution cryo-EM structures in the presence and absence of Ca²⁺, we propose a detailed structural mechanism of the viral disassembly switch. In the presence of Ca²⁺ at acidic pH, tight inter-subunit interactions via Ca²⁺ coordination and carboxyl-carboxyl(ate) interactions between E95 and E97 stabilize the assembly. In these conditions, carboxyl residues are able to bind protons and neutralize their negative charges, which weakens their repulsive force leading to close contact between E95 and E97 (Fig 4). Upon entering the cell, the pH rises and carboxyl groups are deprotonated leading to repulsive forces between them. Putative movement of E97 away from E95 and correlated motion of N98 and N101 destabilize the ion binding site and further promote Ca²⁺ removal in a low Ca²⁺ environment. These concerted conformational rearrangements loosen the stabilizing inter-subunit interactions and ultimately release the switch to disassemble the virus (Movie EV1). The involvement of additional residues such as N101 and N98 in the coordination of the Ca²⁺ binding site suggests a more intricate conformational network responsible for rearrangements beyond the previously postulated carboxylate pair repulsion driving disassembly, which is corroborated by results from a series of mutation experiments [15]. It should be noted that due to the comprehensive nature of conformational changes at the lower radius regions, it is not possible to assign a temporal order to the conformational changes from the two observed structural states of TMV assembly.

Previous studies proposed a critical Ca²⁺ site that interacts with the RNA backbone [5,24]. Such a site could not be located in our two structures (Fig EV3A). In fact, the conformations around the RNA in both our structures resemble what has been referred to as the low Ca²⁺ state [24], with D116, R92, and R90 involved in RNA binding [13] and the direct interaction of R92 and D116. According to our structures, this previously proposed second Ca²⁺ site and the slightly altered conformation at the RNA [5,24] may not be required for TMV stabilization. In addition, residue D109 thought to be important for disassembly was not found to assume different conformations in the two structures and did not form interactions with one of the beforementioned residues (Fig EV3B). To what extent the noted structural differences reflect the different preparation conditions or experimental uncertainties of the previous atomic models is not easy to resolve. Although local resolutions of the here-determined structures at lower

**Table 1. Model validation statistics.**

| Model quantity | Ca²⁺/acidic pH | Water (Model 1) | Water (Model 2) | Water (Model 3) |
|---|---|---|---|---|
| Ramachandran outliers | 0.00% | 0.00% | 0.00% | 0.00% |
| Ramachandran favored | 97.35% | 97.35 | 96.69% | 96.69% |
| Rotamer outliers | 0.00% | 0.00% | 0.00% | 0.00% |
| Clashscore | 1.6 | 3.6 | 4.4 | 4.4 |
| RMS(bonds) | 0.0037 | 0.0051 | 0.0056 | 0.0052 |
| RMS(angles) | 0.63 | 0.76 | 0.82 | 0.82 |
| C-beta deviations | 0 | 0 | 0 | 1 |
| MolProbity score | 1.03 | 1.28 | 1.43 | 1.43 |
| DipCheck chi-score | −0.59 | −0.61 | −0.99 | −0.67 |
| RSCC(mask) | 0.86 | 0.86 | 0.86 | 0.86 |

Statistics for the refined Ca²⁺/acidic pH atomic model and the three different models from TMV in water.

radius drop to ∼ 2.5 Å, maps are sufficiently clear to locate all the mentioned side chains with high confidence (Fig 3). In order to confirm that our batch of TMV presents a biologically active virus, we also showed experimentally that our sample is capable of infecting tobacco plants (Fig EV4).

The proposed structural destabilization mechanism offers the possibility of a cooperative disassembly switch between subunits: the removal of a $Ca^{2+}$ from its coordination site at lower radius has an immediate effect on the neighboring $Ca^{2+}$ sites, which are located in close proximity of 10 Å. In the case of the 5′ and 3′ ends of the virion, the end subunits can only weakly interact at the shared $Ca^{2+}$ coordination site in the $Ca^{2+}$/acidic pH condition as they lack the stabilizing neighboring subunit residues. Upon drop of $Ca^{2+}$ concentration, the ends are even further destabilized and more easily accessible to the pulling replisome machinery. This shared $Ca^{2+}$ site provides a direct explanation of preferred capsid opening at the virion ends and cooperative weakening of RNA-coat protein interactions thus facilitating cotranslational virion disassembly by the replisome [7]. Although the lower radius region of the virion is destabilized in low $Ca^{2+}$ and basic environments, we find that the large part of the CP conformation is not affected by these environmental changes. This is an important aspect of the CP plasticity, which only requires a subtle destabilization of the metastable switch to trigger cotranslational disassembly [26] and, at the same time, to be sufficiently stable to allow for reassembly of the virion after viral replication [7].

# Material and Methods

### Sample preparation

Tobacco mosaic virus sample was isolated as described in ref. [16] and stored in 0.1 M Tris–HCl pH 7.0, 0.02% NaN₃ ($w/v$) at a concentration of ∼ 33 mg/ml at 4°C. A total of 50 μl of virus stock solution was dialyzed for 1 h at room temperature against 50 ml of 0.1 M NaOAc pH 5.2, 20 mM CaCl₂, and 50 ml of MilliQ H₂O, respectively. Before plunge-freezing, sample concentration was adjusted to 22 and 1.1 mg/ml for the $Ca^{2+}$/acidic pH and the water condition, respectively. A total of 3.6 μl were applied on holey carbon grids (C-flat 300 mesh R2/2, Protochips) that had been glow-discharged in an EasyGlow (Pelco) device. Grids were plunge-frozen in liquid ethane using a Vitrobot Mark IV (Thermo Fisher Scientific) with a blotting time of 2 s at 10°C and 100% humidity.

### Electron microscopy

Data acquisition was performed on a Titan Krios microscope (Thermo Fisher Scientific) operated at 300 kV, through a Gatan Quantum 967 LS energy filter using a 20 eV slit width in zero-loss mode. The dataset was recorded on a Gatan K2-Summit direct electron detector operated in super-resolution mode, at a calibrated magnification of 215,000 (resulting in a super-resolution pixel size

**Table 2. RMSD values for model comparisons.**

| | Water (Model 1) | Water (Model 2) | Water (Model 3) |
|---|---|---|---|
| (a) Main and side-chain RMSD between models of TMV (90–110) | | | |
| Ca²⁺/acidic pH | 2.80 Å | 2.53 Å | 2.38 Å |
| Water (Model 1) | – | 0.94 Å | 1.28 Å |
| Water (Model 2) | – | – | 0.75 Å |
| (b) Main and side-chain RMSD between models of TMV Δ(90–110) | | | |
| Ca²⁺/acidic pH | 0.33 Å | 0.40 Å | 0.40 Å |
| Water (Model 1) | – | 0.24 Å | 0.24 Å |
| Water (Model 2) | – | – | 0.01 Å |
| (c) Main-chain RMSD between models of TMV (90–110) | | | |
| Ca²⁺/acidic pH | 2.21 Å | 1.93 Å | 1.92 Å |
| Water (Model 1) | – | 0.43 Å | 0.46 Å |
| Water (Model 2) | – | – | 0.24 Å |
| (d) Main-chain RMSD between models TMV Δ(90–110) | | | |
| Ca²⁺/acidic pH | 0.09 Å | 0.09 Å | 0.09 Å |
| Water (Model 1) | – | 0.01 Å | 0.01 Å |
| Water (Model 2) | – | – | 0.01 Å |

RMSD values (main and side chain combined) for residues 90–110 and Δ(90–110) are shown in (a) and (b). The respective main-chain only RMSD values are shown in (c) and (d).

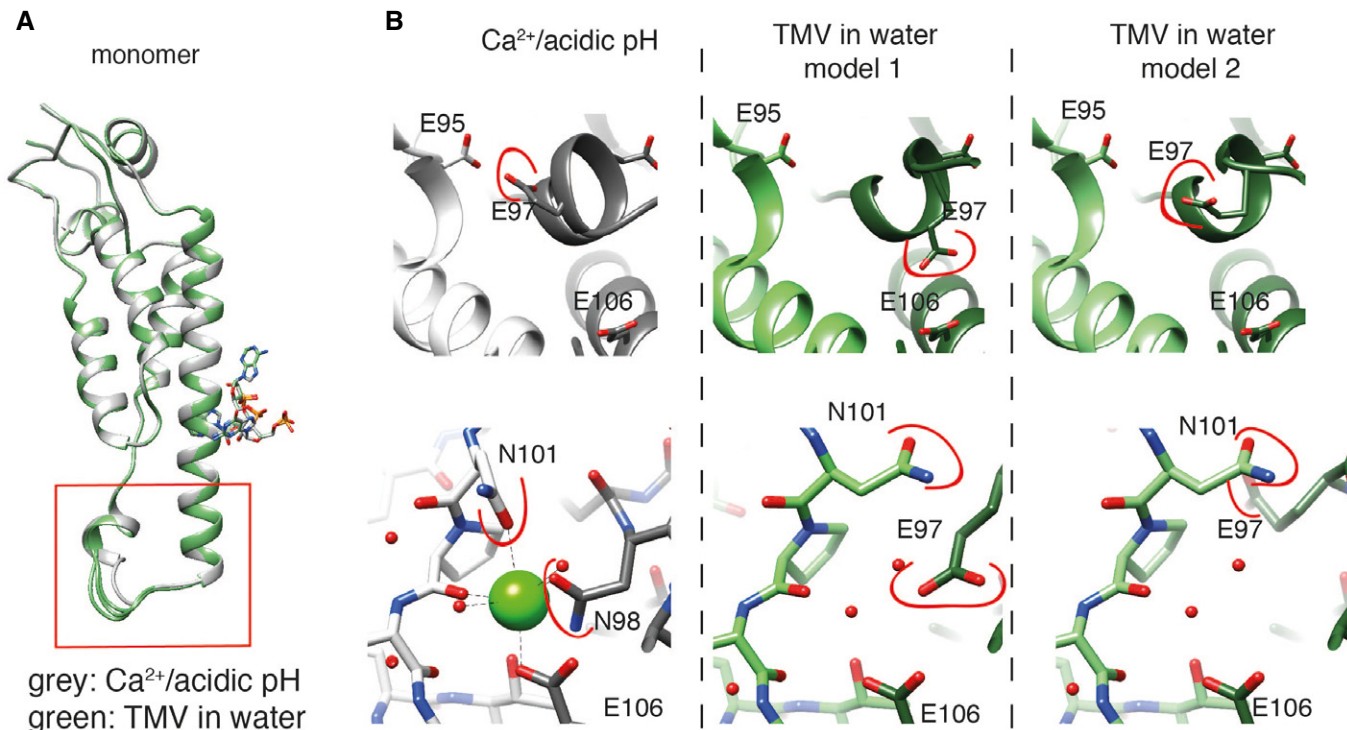

**Figure 3.  Model comparison of Ca²⁺/acidic pH and water structural states.**

A  Superposition of the monomer structures with the lower radius region highlighted in the red box. The Ca²⁺/acidic pH state (gray) shows an additional α-helical segment when compared with the water models (green).

B  Comparison of the E95–E97 interaction (top row) and of the Ca²⁺ binding site in the Ca²⁺/acidic pH model (bottom row). Close proximity of E95 and E97 in the Ca²⁺/acidic pH state, whereas in water, E97 flips toward E106 and is rather flexible. No Ca²⁺ ion and corresponding coordination is evident in the water model. Residues that change conformation are marked in red. Adjacent subunit models are displayed in lighter shade of the main color. Model 3 is not shown due to high structural similarity with model 2 in the displayed region.

of 0.319 Å on the object scale) with a defocus range of 0.15–0.35 μm. For the TMV in water, a total of 20 frames were recorded in movies of 5 s exposure at a dose rate of ~ 2.6 e⁻/physical pix/s, accumulating a total dose of 30.8 e⁻/Å² at the sample level. For the TMV in Ca²⁺/acidic pH conditions, a total of 40 frames were recorded in movies of 4-s exposure with a dose rate of ~ 3.7 e⁻/physical pix/s, accumulating a total dose of 41.3 e⁻/Å² at the sample level. For both samples, data collection was performed on a single grid using SerialEM [27].

**Image processing**

After visual inspection of the micrographs, 62 images for the TMV in water, and 197 images for the TMV in Ca²⁺/acidic pH conditions, were selected and both datasets were processed in the same way. Movie frames were aligned and dose-compensated with MotionCor2 [28] using patch-based alignment (5 × 5) followed by 1/2 cropping in the Fourier domain, resulting in 2× lower pixel sampling and a pixel size of 0.638 Å. Contrast transfer function parameters for the micrographs were estimated using Gctf [29]. Helix coordinates were determined automatically using MicHelixTrace [30], resulting in ~ 20,000 segments for each sample. Complete 2D and 3D classifications and refinements were performed using RELION implementation of single-particle based helical reconstruction [31], including

per-particle refinement of CTF parameters, correction of estimated beam tilt and "Bayesian polishing" [32]. Helical symmetry parameters were refined to a helical rise/rotation of 1.405 Å/22.036° and 1.406 Å/22.038° for the Ca²⁺/acidic and the water structure, respectively. The reported overall resolutions for TMV of 2.0 Å in Ca²⁺/acidic pH and 1.9 Å in water conditions were calculated using the Fourier shell correlation (FSC) 0.143 criterion. The final maps were corrected for the modulation transfer function of the detector and sharpened by applying a negative B factor that was estimated using automated procedures [33] (−41 Å² for the TMV in water and −42 Å² for the TMV in Ca²⁺/acidic pH conditions). Local resolution maps were calculated with BlocRes [34] at a 0.5 FSC cutoff and the maps were subsequently locally filtered. For each local window, we used a hyperbolic tangent low-pass-filter with a fall-off of 0.1 and cutoff frequency given by the local resolution. To annotate significant molecular map features in the 3D reconstruction and to control false positive voxels, confidence maps using local resolution information were generated [22].

**Atomic model building and refinement**

Atomic models were built and refined as 9-mers in order to account for inter-subunit interactions. PDB *4udv* [16] was used as starting model and rigid body fitted into the processed maps using *Chimera*

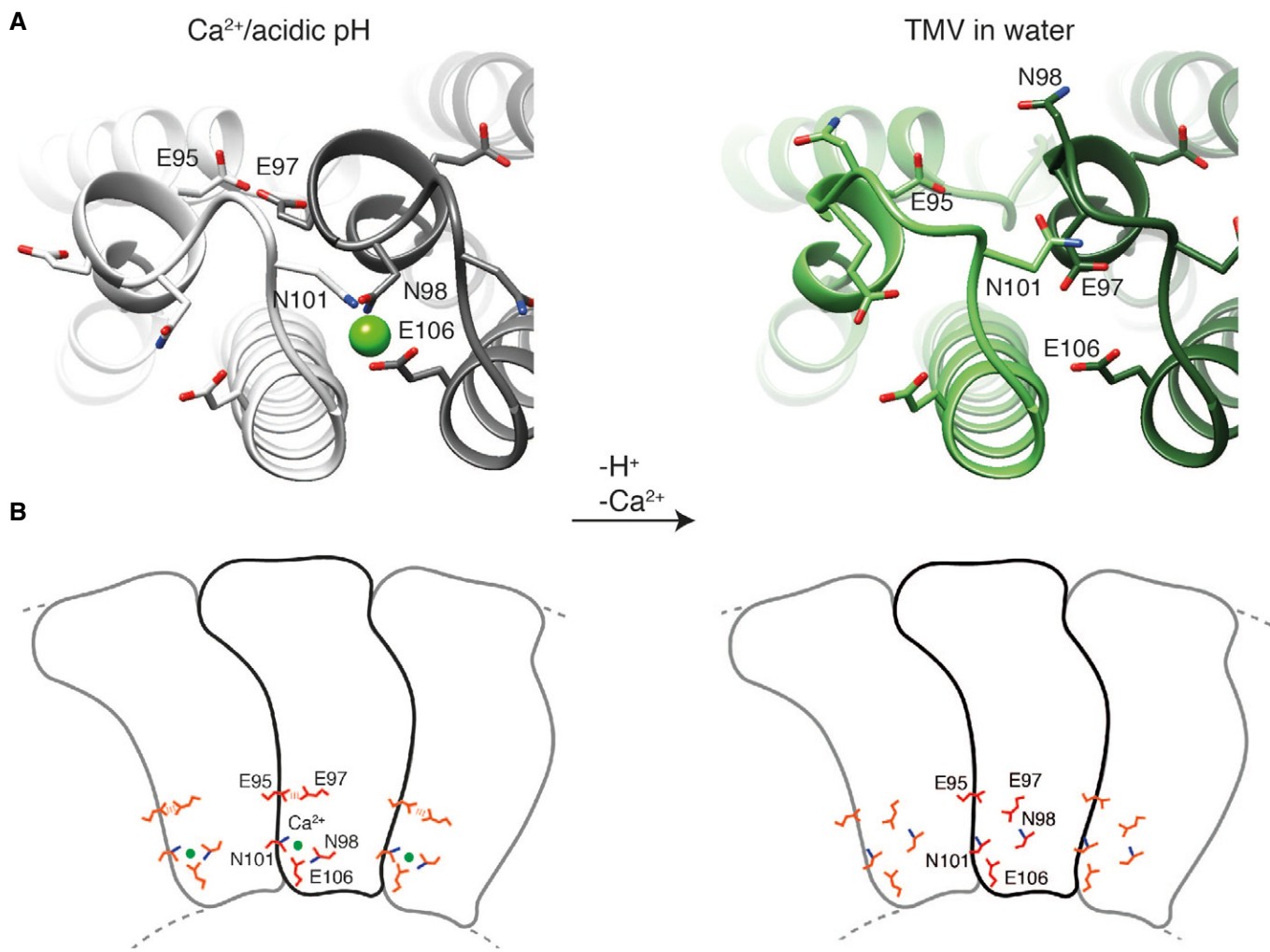

**Figure 4. Toward a disassembly mechanism based on the Ca²⁺/acidic pH and water states.**

A   The close proximity of the E95–E97 interaction and tight coordination of Ca²⁺ in the Ca²⁺/acidic pH state site suggests a mechanism in which, upon pH change and Ca²⁺ removal, repulsive forces between charged carboxylates destabilize the network of interactions at lower radius releasing the switch for viral disassembly.

B   Schematic presentation of coat protein with neighboring subunits including main residues of TMV in the Ca²⁺/acidic pH state (left) and in water (right) responsible for the metastable disassembly switch.

[35]. Additional $H_2O$ and $Mg^{2+}/Ca^{2+}$ ions were placed in the maps where biochemically appropriate and confirmed using the 1% FDR thresholded maps. $Mg^{2+}$ was placed in proximity of the RNA and justified by the known tendency of RNA to be stabilized by $Mg^{2+}$ ions. The $Ca^{2+}$ ion was identified by the combination of high map values, octahedral coordination, and lower B-factors in the respective region (Fig EV5), which distinguishes it clearly from water molecules. The $Ca^{2+}$ ion was only found in the cryo-EM structure with high $Ca^{2+}$ concentrations. Several rounds of real-space refinement with *phenix.real_space_refine* [36] using electron scattering factors and manual rebuilding with *Coot* [37] were done to obtain the presented models. Refinement was performed using rotamer, Ramachandran, and C-β restraints in addition to standard restraints of bond lengths, angles, etc. Real-space refinement was carried out with global minimization and local grid search options activated. Atomic coordinates and B-factors were refined against the

sharpened and locally filtered maps. Residues 154–158 were not modeled as the corresponding map features were not sufficiently well resolved. An additional significant map feature at 1% FDR at the N terminus was modeled as a previously reported N-terminal acetylation [38]. Grouped atomic displacement factors (ADP) were refined with *phenix.real_space_refine.* Validation scores were calculated with *phenix.molprobity* [39], *phenix.em_ringer* [40], and DipCheck [41]. To assess overfitting of the refinement, we introduced random coordinate shifts into the final models using the program *phenix.pdbtools* with the shake option and a mean error of 0.5 Å, followed by refinement against the first unfiltered half-map (half-map 1) with the same parameters as above. Comparisons of FSC curves of the randomized model refined against half-map 1 versus half-map 1 and the FSC curve of the same model map versus half-map 2 do not indicate overfitting (Fig EV1B). Simulated model maps were calculated with *LocScale* [42].

## Infection of *Nicotiana tabacum* with TMV

The Bel B and Bel W3 variants of *N. tabacum* were chosen for infection experiments as they are known to be sensitive to TMV. Seeds distributed on a soil surface were watered and placed in a greenhouse for germination and cultivation at the following growth conditions: length of day: 14 h, day: 28°C/night: 22°C, relative humidity: 70%. Seedlings were piqued after 16 days, repotted after 31 days for the first time, and repotted after 58 days for the second time. Before infection, the plants reached a height of 65–85 cm. On Day 60, one plant of both variants was infected with the TMV whereas the second plant was cultivated free of virus as a control. They were grown under ambient room temperature conditions and 16 h of neon light. For infection, 25 µl of tobacco mosaic virus stock (33 mg/ml) was diluted into 10 ml PBS pH 7.5 and mixed with 105 mg Silicon carbide (SiC, 200–450 mesh, Sigma-Aldrich) in a porcelain mortar. SiC was used as an abrasive to cause small wounds and lesions supporting virus entry [43,44]. The pestle was dipped into the virus/SiC suspension and rubbed gently onto the top surface of each plant leaf [45,46]. Ten days after the infection event, first symptoms of TMV replication were visible [47]. The variety Bel B developed deformed leaves, yellowish spots, and new leaves were unusually light green and showed stunted growth. The variety Bel W3 showed lesions, necrotic spots on the leaves, and stunted growth. After 35 days of infection, the plants possessed heights of 100 and 124 cm of Bel B and of Bel W3, respectively, whereas the corresponding non-infected control plants were of 168 and 167 cm heights.

## Figure preparation

FSC and ADP graphics were visualized with ggplot2 in R [48,49]. Chimera [35] was used for the figure preparation of the molecular maps and atomic models and for preparation of the movie.

# Data availability

The accession numbers for the $Ca^{2+}$ (acidic)/water cryo-EM maps, and corresponding atomic coordinate models are EMD-10130/EMD-10129 (EMDB, www.ebi.ac.uk/pdbe/emdb) and PDB ID 6SAG/PDB ID 6SAE (PDB, www.ebi.ac.uk/pdbe), respectively. The two TMV micrograph sets of the $Ca^{2+}$/acidic and water condition have been deposited to EMPIAR databank and have been assigned accession ID 10306 and 10305, respectively.

**Expanded View** for this article is available online.

## Acknowledgements
We are grateful to Thomas Hoffmann and Jurij Pecar (IT Services) for maintenance of the high-performance computing at EMBL. We thank Wim Hagen and Arjen Jakobi for discussions on the microscope magnification calibration. We would like to thank the Institute of Bio- and Geosciences—Plant Sciences (IBG-2) within the Forschungszentrum Jülich GmbH, especially Beate Uhlig, for providing expertise, material, and space in the greenhouses to grow the tobacco plants.

## Author contributions
FW, MB, and CS designed research. FW prepared cryo-samples, acquired data, and computed 3D reconstructions. MB built and interpreted atomic models. IvdH performed tobacco infection experiments. MB, FW, and CS wrote the article with input from IvdH.

## Conflict of interest
The authors declare that they have no conflict of interest.

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
