## [Review Process File · EMBO Reports]

Elucidation of the viral disassembly switch of tobacco mosaic virus

Felix Weis, Maximilian Beckers, Iris von der Hocht, Carsten Sachse

Review timeline:

Submission date:	9 May 2019
Editorial Decision:	17 June 2019
Revision received:	17 July 2019
Editorial Decision:	5 August 2019
Revision received:	20 August 2019
Accepted:	22 August 2019

Editor: Achim Breiling

Transaction Report:

1st Editorial Decision

17 June 2019

Thank you for the submission of your research manuscript to EMBO reports. We have now received reports from the three referees that were asked to evaluate your study, which can be found at the end of this email.

As you will see, all referees think that the findings are of high interest, but they also have several comments, concerns and suggestions that need to be addressed to render the manuscript suitable for publication in EMBO reports. As the reports are below, I will not further detail them here, also as I feel that all points need to be addressed as indicated by the referees.

Given the constructive referee comments, we would like to invite you to revise your manuscript with the understanding that all referee concerns must be addressed in the revised manuscript and in a detailed point-by-point response. Acceptance of your manuscript will depend on a positive outcome of a second round of review. It is EMBO reports policy to allow a single round of revision only and acceptance or rejection of the manuscript will therefore depend on the completeness of your responses included in the next, final version of the manuscript.

Revised manuscripts should be submitted within three months of a request for revision; they will otherwise be treated as new submissions. Please contact me if a 3-months time frame is not sufficient so that we can discuss the revisions further.

When submitting your revised manuscript, please also carefully review the instructions that follow below. Failure to include requested items will delay the evaluation of your revision. When submitting your revised manuscript, we will require:

- 1) a .docx formatted version of the final manuscript text (including legends for main figures, EV figures and tables), but without the figures included. Please make sure that the changes are highlighted to be clearly visible. Figure legends should be compiled at the end of the manuscript text.

2) individual production quality figure files as .eps, .tif, .jpg (one file per figure), of main figures and EV figures. Please upload these as separate, individual files upon re-submission.

The Expanded View format, which will be displayed in the main HTML of the paper in a collapsible format, has replaced the Supplementary information. You can submit up to 5 images as Expanded View. Please follow the nomenclature Figure EV1, Figure EV2 etc. The figure legend for these should be included in the main manuscript document file in a section called Expanded View Figure Legends after the main Figure Legends section. Additional Supplementary material should be supplied as a single pdf labeled Appendix. The Appendix should have page numbers and needs to include a table of content on the first page (with page numbers) and legends for all content. Please follow the nomenclature Appendix Figure Sx, Appendix Table Sx etc. throughout the text, and also label the figures and tables according to this nomenclature.

For more details please refer to our guide to authors:

<http://embor.embopress.org/authorguide#manuscriptpreparation>

See also our guide for figure preparation:

http://www.embopress.org/sites/default/files/EMBOPress_Figure_Guidelines_061115.pdf

4) a complete author checklist, which you can download from our author guidelines (<http://embor.embopress.org/authorguide#revision>). Please insert page numbers in the checklist to indicate where the requested information can be found in the manuscript. The completed author checklist will also be part of the RPF.

5) that primary datasets produced in this study are deposited in an appropriate public database. See: <http://embor.embopress.org/authorguide#datadeposition>

The accession numbers and database should be listed in a formal "Data Availability " section (placed after Materials & Methods) that follows the model below. Please note that the Data Availability Section is restricted to new primary data that are part of this study.

Data availability

- RNA-Seq data: Gene Expression Omnibus GSE46843

(<https://www.ncbi.nlm.nih.gov/geo/query/acc.cgi?acc=GSE46843>)

- [data type]: [name of the resource] [accession number/identifier/doi] ([URL or identifiers.org/DATABASE:ACCESSION])

6) We strongly encourage the publication of original source data with the aim of making primary data more accessible and transparent to the reader. The source data will be published in a separate source data file online along with the accepted manuscript and will be linked to the relevant figure. If you would like to use this opportunity, please submit the source data (for example scans of entire gels or blots, data points of graphs in an excel sheet, additional images, etc.) of your key experiments together with the revised manuscript. If you want to provide source data, please include size markers for scans of entire gels, label the scans with figure and panel number, and send one PDF file per figure.

7) Our journal encourages inclusion of *data citations in the reference list* to directly cite datasets that were re-used and obtained from public databases. Data citations in the article text are distinct from normal bibliographical citations and should directly link to the database records from which the data can be accessed. In the main text, data citations are formatted as follows: "Data ref: Smith et al, 2001" or "Data ref: NCBI Sequence Read Archive PRJNA342805, 2017". In the Reference list, data citations must be labeled with "[DATASET]". A data reference must provide the database name, accession number/identifiers and a resolvable link to the landing page from which the data can be accessed at the end of the reference. Further instructions are available at: <http://embor.embopress.org/authorguide#referencesformat>

8) Regarding data quantification and statistics, can you please specify, where applicable, the number "n" for how many independent experiments (biological replicates) were performed, the bars and error bars (e.g. SEM, SD) and the test used to calculate p-values in the respective figure legends. Please provide statistical testing where applicable. See: <http://embor.embopress.org/authorguide#statisticalanalysis>

9) Please format the references according to our journal style. See: <http://embor.embopress.org/authorguide#referencesformat>

In addition I would need from you:

- a short, two-sentence summary of the manuscript
- two to four bullet points highlighting the key findings of your study (please remove the key points already provided from the main manuscript text, and submit these together with the synopsis blurb, as a separate word document).
- a schematic summary figure (in jpeg or tiff format with the exact width of 550 pixels and a height of not more than 400 pixels) that can be used as a visual synopsis on our website.

I look forward to seeing a revised version of your manuscript when it is ready. Please let me know if you have questions or comments regarding the revision.

REFEREE REPORTS

Referee #1:

Weiss et al investigate the mechanism of capsid disassembly of TMV. They use high-resolution cryo-EM and single particle helical reconstruction to compare capsid reconstructions in high calcium and in water. The structures revealed a shared calcium binding site between subunits, which has not been observed directly previously. At high calcium concentration, the intact calcium site favors coordination with the so-called "Caspar carboxylates", which are located near the inter-subunit interface. The study reveals this interaction for the first time at near-atomic level. This geometry breaks down at low calcium levels, leading to exposure of the charged carboxylates, whose repulsive forces are proposed to cause disassembly of the structure.

The proposed mechanism is biochemically sound and accurately described. The insights represent a genuine conceptual advance in our understanding of viral disassembly, because the authors are the first to visualize the molecular geometry by direct methods using cryo-EM, which has not been possible at lower resolution. In addition to pushing the limits of helical reconstruction (at 2.0 Angstrom, the presented excellent cryo-EM map has by far the highest spatial resolution of any helical structure deposited in the EM database to date), the authors have used state-of-the-art image processing methods including extensive validation of both map and model. They used a novel algorithm to assess the confidence level of the cryo-EM map (by rendering their maps at 1% "false discovery rate" - a method they have described in detail elsewhere), which represents a more objective way of gauging local map quality. The virus was confirmed active by causing clear symptoms in infected tobacco plants.

The paper is well written, significant, and is suitable for publication in EMBO Reports after minor revisions.

I have only a few comments:

Throughout the paper, the authors refer to the reconstructed electron potential as "density". Although this term is increasingly used both in X-ray crystallography and cryo-EM, it strictly refers to electron density sampled by uncharged X-ray photons. Reconstructions by cryo-EM are a result of sampling the object with a charged electron wave and are thus sensitive to charged side chains such as the carboxylates discussed in the paper. Since the present work is at the forefront of pushing the technical boundaries of cryo-EM and specifically describes charged groups, it may be prudent to correctly refer to the reconstruction as either "the cryo-EM map", "electron (or Coulomb) potential" or simply "the map", and replace the term "density" with "potential" or "map features".

Furthermore, the authors should mention in the methods section, if they have used X-ray scattering factors for their atomic model refinement, or electron scattering factors.

The authors list discrete numbers for ions and water molecules identified in their map. However, there is only a single sentence describing how solvent molecules (water) were assigned ("Additional H₂O and Mg²⁺/Ca²⁺ ions were placed using the 1% FDR thresholded density"). How were they distinguished from Calcium and Magnesium ions and how were charges assigned? If these chemical properties were inferred from geometry or automatically assigned by the molecular refinement program, this should be mentioned. The lower atomic B-factors of Ca-stabilized residues in the low pH/Ca²⁺ condition are indicators for correct assignment of Ca²⁺, as well as the coordination geometry shown in figure 3b. Also, Fig. 1e shows beautiful map features that may convincingly correspond to water molecules in hydrogen-bonding condition with nearby polar side chains. Were all peaks in the 1%FDR map that are not part of the structure assigned to water molecules? What is the RMSD of their distance to hydrogen-bonding partners?

Referee #2:

The manuscript by Weis and co-authors "Elucidation of the viral disassembly switch of tobacco mosaic virus" reports revelation of conformational changes in TMV capsid proteins induced by decreasing Ca²⁺ concentration and raising pH to a neutral level. The authors have obtained two TMV virus structures at higher resolution (~ 2.1 Å) at nearly native conditions and at modified environment. The authors suggested that Ca²⁺ removal and increase in pH destabilize interactions between adjacent subunits that may trigger the viral disassembly. This is an interesting hypothesis and it would be important to discuss it with a much broader audience.

The experiments are interesting and EM structural study has been performed with the high level of expertise. Curiously, the authors did not provide however any quantitative assessments of differences. Unfortunately maps and fitted atomic models of the TMV at different conditions were not available, so it was not possible to verify conformational changes shown in the movie.

The results of this study have risen a few questions:

It remains unclear if the hypotheses is correct: were the authors able to show that concentration of the Ca²⁺ has been increased after viral replication so the new viral progeny of the TMV were able to assemble properly? It seems that something has happen in plants to allow to produce stable fully assembles viruses. How did it happen?

If the concentration of Ca inside the plant cell compared with the extracellular environment does not change then the new virions will not be assembled, otherwise the hypothesis is not quite correct and should be modified.

In the isolation the observed fact is possibly correct, but not related to the reality and to the process of infection.

It is unclear, what was the difference between infections of the Bel B and Bel W3 plant variants. It seems that the EM sample grid preparation was not related to the type of infected plants. The link between the experimental results on plants and EM structural studies is absent. Were these plants

used just to get the TMV virions? If so, that has to be stated from the very beginning. As it follows from the MS the conditions for the viruses were changed in a tube before grid preparations for EM structural analysis. What was the point to have two different plant species? Possibly would be more useful to show by EM that specific mutants that are not able to bind Ca^{2+} and, therefore, are not infective.

The authors have spent a lot of efforts to convince a reader that the calibration of the EM (a magnification) was perfectly done. What was the accuracy of these assessments? The error in magnification of $\sim 1\%$ is rather common, so it is interesting to know that the resolution dropped down if the rise of capsid subunits was about 1% less compared to the previous studies. Was it of any significance? How was it proved?

The authors are mentioning that they have used some interesting (published by the senior author) approaches to do sharpening EM maps and to represent EM densities for the tracing polypeptide chains. Nonetheless, very little information was provided on which parameters have been used and how the local filtering has been done, which criteria were used? Is a representation of the filter profile in Fourier space available?

For the majority of readers the sentence "To annotate significant molecular density in the 3D reconstruction and to control false positive voxels, confidence maps(30) were generated, where local resolution information was incorporated to increase statistical power, i.e. to decrease the amount of false negative voxels" will be absolutely mysterious (Page 6). The authors have to explain what is the difference between "false positive voxels" and "false negative voxels". How the local resolution information may increase statistical power (??) of what?

It is unclear how the water molecules were identified within the maps, how locations of the real water molecules were identified and distinguished from the "false" positive maxima?

The authors did not show maps and did not provide any numerical information on how well the atomic models corresponds to the maps. It is unclear how the presence of the additional α -helical segment in the Ca^{2+} /acidic pH model corresponds to the EM densities. The local bits of maps with fitted models should be shown without any manipulations with the maps. What does say to us comparison of two models of the TMV structure in "water". It would be useful to see a difference map and local resolution maps of these areas in all conformations. What was an RMSD between these two models of these conformations? The maps of confidence look strange: distorted, disruptive and noisy. Are the authors sure that this is the best way to use this methods for the interpretation of maps? The shown fits do not look convincing.

Minor comments:

The authors have to indicate in the figure one where are the "lower", "medium", and "high" radii, what sizes they are? The authors show show sizes in figures (not the scale bars, but sizes).

Figure 1, a. These images are so small that it is impossible to see what the authors want to show. Put it as a first supplementary figure and at bigger scale. The diffraction patters are so small: what do the authors want to show to a reader? It is nearly impossible to see neither Thon rings, nor any reflections related to the TMV. If the authors want to show some differences they have to indicate them on figures.

Figure 2. The legend does not sound right,; it is confusing what is "top" and "bottom " rows, Possibly the top an bottom have to correspond to the left and right?

Correct "Vitrobot mark IV" for "Vitrobot Mark IV"

Please explain why the grids were frozen at $10\text{ }^{\circ}\text{C}$ but not at room temperature or $4\text{ }^{\circ}\text{C}$?

What was a reason to collect data at super resolution mode, if the data were then coarsened?

The meaning of the word "cropping" is not the same as "coarsening": "...Fourier

space cropping (by a factor 2)". Please rephrase and give some explanations.

Please explain: "Real-space refinement was carried out with global minimization together with a local grid search. "

"...we introduced random coordinate shifts into the final models...". Was it used a Gaussian distribution? Parameters? Provide more information and parameters, how it was done.

"Both samples were plunge-frozen and imaged using a 300kV electron microscope equipped with a direct electron detector..." Did the authors meant the K2 camera? Possibly would be better to be more explicit.

It is a bit strange sentence: ".. whereas in water it is replaced by H₂O molecules." Possibly some rephrasing is needed. According to the general knowledge: water = H₂O?

Referee #3:

Weis et al. represent two structures of Tobacco mosaic virus at some 2 Å resolution, derived from electron cryo microscopy. The two structures represent the conformational states at low pH in the presence of Ca²⁺ and in water. The thorough comparison of the structures identifies the importance of Ca²⁺-ions for the stabilization of the inter-subunit interaction. For this, reliably resolving the side-chain conformations of the individual residues at the inter-subunit interface together with individual Ca²⁺-ions and the water-molecules is essential.

The authors have done a methodological sound, structural work, in which they make sure that the detected differences in the structural models are based on statistically significant observations. There rigorous analysis is rewarded with the so far best resolved maps of Tobacco Mosaic Virus, which do not only shed light on a viral uncoating mechanism but are also interesting to the EM-community.

The manuscript should be further improved by:

- 1) Showing that the densities attributed to Ca²⁺-ions are not water or other ions, because large parts of the discussion are based on this assignment. Here it would be sufficient to explain the rationale of the assignment.
- 2) Demonstrating that the observed differences in the structural model coincide with the major differences between the two EM-maps.

Minor issues:

1. Page 4 „3.3Å by electron cryo-microscopy"

There are more recent cryo-EM maps at 2.3 Å (EMD4413; Song, B., et al. (2019) *Ultramicroscopy* 203: 145-154.)

2. Page 19 Figure 2 legend " Detailed map comparison at lower-radius region of Ca²⁺/acidic pH (grey, top row) with the water condition (green, bottom row)."

This should read left and right instead top and bottom

3. The authors refer to "radius" to specify a position of a certain region in respect to the virus. However, the meaning of radius is not intuitively clear. Maybe the authors should once introduce the phrase (e.g. distance from the helical axis) or rephrase altogether. It would also be more precise to quote the distance range rather than referring to "lower" or "higher" radius.

Response to Referee 1:

1. Throughout the paper, the authors refer to the reconstructed electron potential as "density". Although this term is increasingly used both in X-ray crystallography and cryo-EM, it strictly refers to electron density sampled by uncharged X-ray photons. Reconstructions by cryo-EM are a result of sampling the object with a charged electron wave and are thus sensitive to charged side chains such as the carboxylates discussed in the paper. Since the present work is at the forefront of pushing the technical boundaries of cryo-EM and specifically describes charged groups, it may be prudent to correctly refer to the reconstruction as either "the cryo-EM map", "electron (or Coulomb) potential" or simply "the map", and replace the term "density" with "potential" or "map features".

We agree that usage of the word density for cryo-EM maps could be misleading. We used these terms interchangeably as density is being used increasingly by the cryo-EM community. Based on Referee 1's recommendation we replaced the term density by cryo-EM map throughout the manuscript.

2. Furthermore, the authors should mention in the methods section, if they have used X-ray scattering factors for their atomic model refinement, or electron scattering factors.

We used electron scattering factors and mention this now in the text on page 7:

Several rounds of real-space refinement with phenix.real_space_refine[35] using electron scattering factors and manual rebuilding with Coot[36] were done to obtain the presented models.

3. The authors list discrete numbers for ions and water molecules identified in their map. However, there is only a single sentence describing how solvent molecules (water) were assigned ("Additional H₂O and Mg²⁺/Ca²⁺ ions were placed using the 1% FDR thresholded density"). How were they distinguished from Calcium and Magnesium ions and how were charges assigned? If these chemical properties were inferred from geometry or automatically assigned by the molecular refinement program, this should be mentioned. The lower atomic B-factors of Ca-stabilized residues in the low pH/Ca²⁺ condition are indicators for correct assignment of Ca²⁺, as well as the coordination geometry shown in figure 3b. Also, Fig. 1e shows beautiful map features that may convincingly correspond to water molecules in hydrogen-bonding condition with nearby polar side chains. Were all peaks in the 1%FDR map that are not part of the structure assigned to water molecules? What is the RMSD of their distance to hydrogen-bonding partners?

We agree that this important topic may not have been adequately covered in the previous version of the manuscript. Not every peak at 1% FDR was simply assigned to a water molecule. Waters were placed by using the combined information from possible interaction partners, map intensities and confidence maps. We rephrased the sentence and added the following paragraph to the Methods section on page 6:

Additional H₂O and Mg²⁺/Ca²⁺ ions were placed in the maps where biochemically appropriate and confirmed using the 1% FDR thresholded maps. Mg²⁺ was placed in proximity of the RNA and justified by the known tendency of RNA to be stabilized by Mg²⁺ ions. The Ca²⁺ ion was identified by the combination of high map intensities, octahedral coordination and lower B-factors in the respective region (Figure EV5), which distinguishes it clearly from water molecules. The Ca²⁺ ion was only found in the cryo-EM structure with high Ca²⁺ concentrations.

Moreover, we added a histogram of the observed distances between donor and acceptor of hydrogen bonds in **Figure EV2d**. The histogram corresponds well with the observed range of bond lengths described by Jeffrey, George A., An introduction to hydrogen bonding, Oxford University Press, 1997.

Response to Referee 2:

1. It remains unclear if the hypotheses is correct: were the authors able to show that concentration of the Ca²⁺ has been increased after viral replication so the new viral progeny of the TMV were able to assemble properly? It seems that something has happen in plants to allow to produce stable fully assembles viruses. How did it happen?

We believe that this point arises from a misunderstanding as we did not formulate the hypothesis that higher Ca²⁺ concentration can re-assemble the virus inside the cell, we rather carefully discuss this as a potential consequence of our results in the context of plants. In the previous version of the

manuscript, we had been very cautious in separating the presented Results from the Discussion. Results: the cryo-EM structures were determined in buffer conditions mimicking the ionic concentrations of extracellular fluids (high Ca^{2+} /low pH) and lower Ca^{2+} (high pH) inside the cell. This principal buffer relationship was established in early assembly/disassembly studies from Durham et al. 1977 (Virology). Our study does not attempt to demonstrate the Ca^{2+} increase after viral replication and thereby priming re-assembly. Instead we observe that also in water a large part of the coat protein structure is unaffected by the loss of Ca^{2+} and is sufficiently stable to be assembled in a helical virus. Single molecule force spectroscopy by Liu *et al.* (*Biophysical Journal* (2013) more recently established that reassembly is still possible at pH 7.0 and low Ca^{2+} albeit at lower efficiencies.

To avoid any misunderstanding, we reformatted the manuscript with headings and now have a clearly separated Discussion section from the Results. Finally, we say in the Discussion: *Although the lower radius region of the virion is destabilized in low Ca^{2+} and basic environments, we find that the large part of the CP conformation is not affected by these environmental changes. This is an important aspect of the CP plasticity, which only requires a subtle destabilization of the metastable switch to trigger cotranslational disassembly[48] and, ...*

In order to clarify the perhaps misleading statement in the last sentence, we state more carefully in the revised version of the manuscript:

... at the same time, to be sufficiently stable to allow re-assembly of the virion after viral replication[7].

2. It is unclear, what was the difference between infections of the Bel B and Bel W3 plant variants. It seems that the EM sample grid preparation was not related to the type of infected plants. The link between the experimental results on plants and EM structural studies is absent.

A) Were these plants used just to get the TMV virions? If so, that has to be stated from the very beginning. As it follows from the MS the conditions for the viruses were changed in a tube before grid preparations for EM structural analysis. What was the point to have two different plant species?

The major purpose to include the infectivity assays with BelB as well as Bell W3 plant variants was to demonstrate that we are working with virus that is still capable of infecting various tobacco plants (stated in Results on page 12). This is an important point as TMV is a very stable EM specimen, which could be old and therefore potentially inactive regarding its biological function. This aspect has often been overlooked in previous cryo-EM structures of TMV. We did not use the presented infection results for virus production.

B) Possibly would be more useful to show by EM that specific mutants that are not able to bind Ca^{2+} and, therefore, are not infective.

The intriguing suggestion by Referee 2 has already been demonstrated by Lu et al. 1996, which is cited in the Discussion section of our manuscript on page 11. The authors performed co-translational and infectivity assays for D77N, E55Q, E106Q, E95Q, E97Q, D109N and D116N. Only in combination with E95/E97/D109N the infectivity drops to 20 %. To request another series of cryo-EM structures determined from mutant assemblies we consider beyond the scope of this manuscript.

3. The authors have spent a lot of efforts to convince a reader that the calibration of the EM (a magnification) was perfectly done. What was the accuracy of these assessments? The error in magnification of ~ 1% is rather common, so it is interesting to know that the resolution dropped down if the rise of capsid subunits was about 1% less compared to the previous studies. Was it of any significance? How was it proved?

The respective paragraph is now obsolete and was removed as the calibrated pixel size was not correct (first paragraph of response to Referees). The obtained helical parameters using the correct pixel size are now in very good agreement with previously determined values.

4. The authors are mentioning that they have used some interesting (published by the senior author) approaches to do sharpening EM maps and to represent EM densities for the tracing polypeptide chains. Nonetheless, very little information was provided on a) which parameters have been used and b) how the local filtering has been done, which criteria were used? Is a representation of the filter profile in Fourier space available?

a) To clarify the procedure of map generation, we now state in the Methods section on page 6: *To annotate significant molecular map features in the 3D reconstruction and to control false positive voxels, confidence maps using the local resolution information were generated(31). Background noise was estimated outside the particle at the default locations and with sizes of the noise boxes of 100 pixels for both structures.*

For more detailed parameters, we include the figure above for the Referee, which illustrates the regions of the sharpened TMV map used for noise estimation. We also include local noise estimation based on local filtering described (point 4b below).

b) We added a more detailed description of the local filtering and modified the respective part in the Methods section on page 6 accordingly:

Local resolution maps were calculated with BlocRes(30) at a 0.5 FSC cutoff and the maps were subsequently locally filtered as described(30). For each local window we used a hyperbolic tangent low pass filter with a fall-off of 0.1 and cutoff frequency given by the local resolution.

Further details of the procedure can be found in the Confidence map paper (Beckers et al., 2019 *JUUCr Journal*)

5. For the majority of readers the sentence "To annotate significant molecular density in the 3D reconstruction and to control false positive voxels, confidence maps(30) were generated, where local resolution information was incorporated to increase statistical power, i.e. to decrease the amount of false negative voxels" will be absolutely mysterious (Page 6). The authors have to explain what is the difference between "false positive voxels" and "false negative voxels". How the local resolution information may increase statistical power (??) of what?

We agree that this sentence is very technical. We shortened the paragraph on page 6 and refer to the respective paper for details:

To annotate significant molecular map features in the 3D reconstruction and to control false positive voxels, confidence maps using the local resolution information were generated(31).

6. It is unclear how the water molecules were identified within the maps, how locations of the real water molecules were identified and distinguished from the "false" positive maxima?

Thank you for this comment. Referee 1 raised a similar concern. See our response to Referee 1, point 3.

7.

A) The authors did not show maps and did not provide any numerical information on how well the atomic models corresponds to the maps.

In the previous version of the manuscript, we showed model-map FSC curves in **Figure EV1**. Based on the request, we added an additional value of real-space cross-correlation to the validation statistics in **Table 1** to further assess the fit of the model to the map. For the Referee, we provide a residue-specific cross-correlation plots to demonstrate the fit of the individual residues:

The drop in cross-correlation for the water models can be explained by model increase in flexibility in that region as the map intensity also drops.

We also provide all maps and models used for the interpretation to the Referees at the following location: <https://oc.embl.de/index.php/s/IqNgxImlFPZIB8I>

B) It is unclear how the presence of the additional α -helical segment in the Ca^{2+} /acidic pH model corresponds to the EM densities.

For the Referee, we provide a snapshot of the cryo-EM map this region to show the extension of the helical segment by one residue. Please note, the atomic coordinate refinement did not include any secondary structure restraints to enforce helicity in that region.

For clarification, we now state more precisely in the manuscript on page 6: *Comparison of the refined atomic models at lower radius revealed an extended helix of the short α -helical segment in the Ca^{2+} /acidic pH model by the single residue N98 (Figure 3a).*

C) The local bits of maps with fitted models should be shown without any manipulations with the maps. What does say to us comparison of two models of the TMV structure in "water". It would be useful to see a difference map and local resolution maps of these areas in all conformations. What was an RMSD between these two models of these conformations?

In order to exclude any suspected "map manipulation" issues, we provide a difference analysis from unsharpened maps for the Referee as requested (below). The analysis shows that the differences are

so large that they can also be easily visualized in the unsharpened maps, ie. at significantly lower resolution than the map's resolution. The more detailed analysis including conformational differences can be better obtained from the sharpened densities. Therefore, we added an additional figure (**Figure EV5**) showing the comparison between the two sharpened maps of the Ca^{2+} /acidic and water condition. In addition, we also add the requested RMSD between the different models including explicitly the loop region of conformational differences (**Table 2**).

The following paragraph has been added to the Results section on page 9:

*As expected, the structural differences between the Ca^{2+} /acidic pH and water structures in the region outside the 90-110 region are very low with RMSDs below 0.5 Å whereas higher between 2.4 and 2.8 Å inside the lower radius region. The multiple models placed for the water structure in the lower radius region deviate to a smaller but significant extent around 1.0 Å (**Table 2**).*

*In order to confirm the observed differences on the level of the cryo-EM maps, we compared the respective loop regions by difference mapping (**Figure EV5a**). The difference map between the two conditions shows the rearrangement of the α -helical segment including the presence and coordination of the Ca^{2+} ion (**Figure EV5b**). ... Moreover, local resolutions plots of the cryo-EM density justify the placement of side chains and show once more the stabilizing effect of the Ca^{2+} /acidic pH condition (**Figure EV5d**).*

D) The maps of confidence look strange: distorted, disruptive and noisy. Are the authors sure that this is the best way to use this methods for the interpretation of maps? The shown fits do not look convincing.

The advantages and properties of the displayed Confidence maps have been discussed in detail elsewhere (Beckers et al., 2019 *JUCr Journal*). Briefly, it is important to note that Confidence maps are not EM maps: they contain estimated probabilities (false discovery rates) and due to a clear separation of signal from background noise they appear quite sharp in surface rendering displays whereas EM maps have shallow gradients and appear smoother. Especially for the assignment of weaker and more diffuse signal, (i.e. in the examined loop region with lower resolutions and higher flexibility), however, this clear separation and validation by statistical significance using Confidence maps was essential to guide our interpretation of the different conformations. The used Confidence maps are available for inspection at the the following location: <https://oc.embl.de/index.php/s/lqNgxImlFPZIB8l>

For comparison, we also added the sharpened EM densities in **Figure EV5** to the manuscript.

Minor comments:

8. The authors have to indicate in the figure one where are the "lower", "medium", and "high" radii, what sizes they are? The authors show show sizes in figures (not the scale bars, but sizes).

Based on the comment of the Referee 2, we added a definition to the Introduction paragraph at the beginning of the manuscript and also added a scale bar to Figure 1 b.

The following sentence has been added to the Introduction on page 3:

The TMV coat protein folds into a four-helix bundle that is commonly divided in three regions referring to the radial distance from the helical axis [13]: lower radius 20 – 40 Å, middle radius 40 – 60 Å and higher radius 60 – 90 Å.

9. Figure 1, a. These images are so small that it is impossible to see what the authors want to show. Put it as a first supplementary figure and at bigger scale. The diffraction patterns are so small: what do the authors want to show to a reader? It is nearly impossible to see neither Thon rings, nor any reflections related to the TMV. If the authors want to show some differences they have to indicate them on figures.

We increased the size of the micrographs in Fig. 1a and removed the power spectra as suggested.

10. Figure 2. The legend does not sound right,; it is confusing what is "top" and "bottom " rows, Possibly the top an bottom have to correspond to the left and right?

Changed as suggested.

11. Correct "Vitrobot mark IV" for "Vitrobot Mark IV"

Changed as suggested.

12. Please explain why the grids were frozen at 10{degree sign} C but not at room temperature or 4{degree sign} C?

There are technical reasons why we chose to use 10 deg C. On Vitrobot devices, the stabilization of humidity is faster and more reproducible at 10 degrees in comparison with 4 degrees.

13. What was a reason to collect data at super resolution mode, if the data were then coarsened?

For the motion correction with alignment of the movie frames it has advantages for the involved interpolations to use super-resolution images. This is the recommended procedure in the MotionCor2 paper (Zheng et al., Nat Methods, 2017):

“To minimize the attenuation of high-resolution signals due to interpolation, bilinear interpolation is performed on super-resolution pixels.”

The meaning of the word “cropping” is not the same as “coarsening”: “...Fourier space cropping (by a factor 2)”. Please rephrase and give some explanations.

To clarify, the sentence was rephrased in the Methods section on page 5:

Movie frames were aligned and dose-compensated with MotionCor2 (23) using patch-based alignment (5 x 5) followed by 1/2 cropping in the Fourier domain, resulting in 2x lower pixel sampling and a pixel size of 0.638 Å.

This is the recommended routine from MotionCor2. Please, refer to the paper (Zheng et al., Nat Methods, 2017):

“The final image is obtained by cropping the corrected sum in the Fourier domain to the user specified resolution.”

Please explain: "Real-space refinement was carried out with global minimization together with a local grid search. "

Refinement of atomic coordinates aims at optimizing the fit to the map while maintaining a meaningful protein geometry. Any optimization problem can be formulated as minimization problem by changing the sign of the loss function. Finding the global minimum is not straightforward as local optimization algorithms will only converge to local minima, which could be far away from the global minimum. Therefore, global optimization algorithms aim to find better

minima by smart heuristics. Local grid search is an additional option in *phenix.real_space_refine*, which allows to fix rotamer outliers and bad map-model fit. A complete description of the procedure is beyond the scope of our paper and we refer to the available articles of PHENIX (Adams et al., 2010). To make it more clear that these are options to be chosen in *phenix.real_space_refine*, we rephrased the sentence more clearly in the Methods section on page 7:

Real-space refinement was carried out with global minimization and local grid search options activated.

"...we introduced random coordinate shifts into the final models...". Was it used a Gaussian distribution? Parameters? Provide more information and parameters, how it was done.

PHENIX does not give any specifications about the distribution that is used. However, we further specified the actual parameters in the Methods section on page 7:

To assess overfitting of the refinement, we introduced random coordinate shifts into the final models using the program phenix.pdbtools with the shake option and a mean error of 0.5 Å.

"Both samples were plunge-frozen and imaged using a 300kV electron microscope equipped with a direct electron detector..." Did the authors meant the K2 camera? Possibly would be better to be more explicit.

We added the actual specification to the respective sentence in the Results section on page 8:

Both samples were plunge-frozen and imaged using a 300kV electron microscope equipped with a GIF Quantum K2 camera (Figure 1a).

It is a bit strange sentence: ".. whereas in water it is replaced by H2O molecules." Possibly some rephrasing is needed. According to the general knowledge: water = H2O?

We modified the sentence in the Results section on page 10:

Therefore, we conclude that under Ca²⁺/acidic pH conditions this subunit interface is stabilized by a Ca²⁺ ion, whereas in TMV in water condition the respective site is occupied by water molecules.

Response to Referee 3:

The manuscript should be further improved by:

1) Showing that the densities attributed to Ca²⁺-ions are not water or other ions, because large parts of the discussion are based on this assignment. Here it would be sufficient to explain the rational of the assignment.

Thank you for this comment. Referee 1 and 2 raised a similar concern. See our response to Referee 1, point 3.

2) Demonstrating that the observed differences in the structural model coincide with the major differences between the two EM-maps.

Thank you for this comment. Referee 2 raised a similar concern. See our response to Referee 2, point 7C.

Minor issues:

1. Page 4 „3.3Å by electron cryo-microscopy"

There are more recent cryo-EM maps at 2.3 Å (EMD4413; Song, B., et al. (2019) Ultramicroscopy 203: 145-154.)

Thank you for pointing out this very recent structure. We added the respective reference to the Introduction on page 4:

Although TMV was subject to a plethora of structural studies(15–18), resolution of the helical rod was limited to 2.9Å when determined by early X-ray fiber diffraction(5) or later to 3.3 Å by electron cryo-microscopy (cryo-EM) studies(13, 16) and more recently up to 2.3 Å resolution(19).

2. Page 19 Figure 2 legend " Detailed map comparison at lower-radius region of Ca²⁺/acidic pH (grey, top row) with the water condition (green, bottom row)."

This should read left and right instead top and bottom

Changed as suggested (see point 10 Referee 2).

3. The authors refer to "radius" to specify a position of a certain region in respect to the virus. However, the meaning of radius is not intuitively clear. Maybe the authors should once introduce the phrase (e.g. distance from the helical axis) or rephrase altogether. It would also be more precise to quote the distance range rather than referring to "lower" or "higher" radius.

Changed as suggested (see point 8 Referee 2)

2nd Editorial Decision

5 August 2019

Thank you for the submission of your revised manuscript to our editorial offices. We have now received the reports from the three referees that were asked to re-evaluate your study, you will find below. As you will see, all referees now support the publication of your manuscript in EMBO reports. Referee #2 has some further requests and suggestions we ask you to address in a final revised version of your manuscript. As indicated by referee #2, please try to render the manuscript generally more comprehensible for non-expert readers.

Further, I have these editorial requests:

- Please provide the abstract written in present tense.
- Please remove the bullet points and the synopsis blurb from the manuscript file. I have saved these separately, and will forward them to our publisher after acceptance of the manuscript.
- We would like to publish your paper as Scientific Report. Thus, we need to ask you to combine the Results and Discussion sections into one section termed 'Results & Discussion'. See: <http://www.embopress.org/page/journal/14693178/authorguide#researcharticleguide>
- Please format the references according to our journal style. Presently, many references are lacking issue information and page numbers. See also: <http://www.embopress.org/page/journal/14693178/authorguide#referencesformat>
- The labelling of the axes in Figures EV2d and EV5e is too small, and will not be readable in the online version of the figure. Please provide these panels with bigger fonts.
- Please add legends explaining the content to Tables 1 and 2.
- Please add the necessary information for the data availability section, including all IDs, the name of the database and a link to the database.
- Please remove the legend of Movie EV1 from the main manuscript text. Please provide this as separate text file (Movie EV1. Structural transition of the determined Ca²⁺/acidic pH and water state), and ZIP this together with the movie file. Then please upload the combined folder.
- Finally, please find attached a word file of the manuscript text (provided by our publisher) with changes we ask you to include in your final manuscript text, and some queries, we ask you to address. Please provide your final manuscript file with track changes, in order that we can see the modifications done.

REFeree REPORTS

Referee #1:

I am very happy with the revised version. The authors have comprehensively addressed all of my concerns about the initial manuscript.

This groundbreaking work represents a new benchmark for the state of the art in cryo-EM of helical objects, will certainly be cited extensively and should be published as-is without further delay. MW.

Referee #2:

Weiss and co-authors improved the MS significantly, they have addressed nearly all points raised by the reviewers. The differences between the maps were shown nicely. However, there are some minor issues that have to be addressed:

- References are absolutely in a chaotic format.
- Some corrections did not consider the fact that a reader does not know many mathematical details and did not read the previous papers of the senior author.

Additional comments:

1. "We identified a total of 92 water molecules for TMV in water, 71 water molecules under Ca²⁺/acidic pH, whose modelled donor acceptor hydrogen bond lengths correspond to expected reference values (Figure EV2d). In addition, we found 4 Mg²⁺-ions bound to RNA as well as well defined side-chain conformers per CP in both conditions."

That still has to be quantified: how the authors are sure that they have found 92 and 71 water molecules and that was not noise? What was used as reference values?

2. "we were able to consistently analyze the CP map"
 Linguistic issue: can be scientific analysis not consistent?

3. "the lower radius region differs significantly..."
 Please provide in the text the level of significance.

4. "Figure EV2b) revealed that previous studies only poorly resolved this part of the protein as the map values were much weaker than other parts of the protein."
 It is understandable, that the authors were following the request of the first reviewer, however it did not make the meaning of the sentence more clear. Please rephrase the sentence, it sounds confusing "...as the map values were much weaker than other parts of the protein.."
 "...the map values..." -> it is not understandable what do you mean if you compare with parts of the protein. It is a linguistic problem -> electron density?

5. "... regions revealed that the protein backbone follows an alternative path."
 Please explain the difference between paths, what does it mean "alternative", what was the difference.

6. "The multiple models placed for the water structure in the lower radius region deviate to a smaller but significant extent around 1.0 Å"
 Was it only in specific regions? Possibly will be useful to indicate them by numbering aa?

7. "Annotation of the Ca²⁺ ion is based on the strong map intensities at the respective site and octahedral coordination"
 Possibly it did not help to follow recommendations of the first reviewer "map intensities" -> electron densities?

8. "... be noted that due to the convoluted nature of conformational changes at the lower radius regions"
 What is it "...the convolution nature...of changes". Please have respect for an average reader.

9. "For each local window we used a hyperbolic tangent low pass filter with a fall-off of 0.1 and

cutoff frequency given by the local resolution."

It is an interesting filter, could you provide a profile with parameters on it? Again, please consider an average reader.

10. "Mg²⁺ was placed in proximity of the RNA and justified by the known tendency of RNA to be stabilized by Mg²⁺ ions. The Ca²⁺ ion was identified by the combination of high map intensities, octahedral coordination and lower B factors in the respective region (Figure EV5), which distinguishes it clearly from water molecules. The Ca²⁺ ion was only found in the cryo-EM structure with high Ca²⁺ concentrations."

This is a very useful piece of information. How much was the difference in the local map densities corresponding to water and Mg or Ca ions. How was it assessed?

Referee #3:

All my previous concerns have been adequately addressed.

2nd Revision - authors' response

20 August 2019

Response to Comments of Referee #2:

Weiss and co-authors improved the MS significantly, they have addressed nearly all points raised by the reviewers. The differences between the maps were shown nicely.

We thank Referee #2 for his/her overall positive assessment of the manuscript.

However, there are some minor issues that have to be addressed:

References are absolutely in a chaotic format.

Unfortunately, synchronization of the reference manager failed in the submitted document. We fixed this issue in the revised version of the manuscript.

Some corrections did not consider the fact that a reader does not know many mathematical details and did not read the previous papers of the senior author.

By addressing the following comments, we took the extra effort to make the respective statements more comprehensible and also added some additional clarifications.

Additional comments:

1. "We identified a total of 92 water molecules for TMV in water, 71 water molecules under Ca²⁺/acidic pH, whose modelled donor acceptor hydrogen bond lengths correspond to expected reference values (Figure EV2d). In addition, we found 4 Mg²⁺-ions bound to RNA as well as well defined side-chain conformers per CP in both conditions."

That still has to be quantified: how the authors are sure that they have found 92 and 71 water molecules and that was not noise? What was used as reference values?

For reference, the technical explanation to the process we did already add to the Methods section on page 11:

Additional H₂O and Mg²⁺/Ca²⁺ ions were placed in the maps where biochemically appropriate and confirmed using the 1% FDR thresholded maps. Mg²⁺ was placed in proximity of the RNA and justified by the known tendency of RNA to be stabilized by Mg²⁺ ions. The Ca²⁺ ion was identified by the combination of high map values, octahedral coordination and lower B-factors in the respective region (Figure EV5), which distinguishes it clearly from water molecules. The Ca²⁺ ion was only found in the cryo-EM structure with high Ca²⁺ concentrations.

Background noise is controlled due to the application of confidence maps that were interpreted at a false discovery rate of 1%. However, it is correct, we cannot completely exclude that some of the

modelled waters are noise. Therefore, we rephrased the respective sentences and exchanged “identified” with “modelled” in the Results section, to make the process of molecule or ion placement clearer. The requested reference is: Jeffrey, George A.; *An introduction to hydrogen bonding*, Oxford University Press, 1997

We added these statements to the Results section:

The map can also be used to locate non-protein components such as water molecules and metal ions. In order to minimize the influence of noise during molecule and ion placements, we used recently developed confidence maps at a 1 % false discovery rate threshold (FDR)[22] that is known to suppress noise in comparison with fixed sigma thresholds of EM maps. Using these confidence maps together with expected donor-acceptor hydrogen bond lengths[23] (Figure EV2d), we modelled a total of...

Due to the proximity to RNA, we modelled as 4 Mg²⁺-ions bound to RNA as well as well-defined side-chain conformers per CP in both conditions (Figure 1c, d, e).

2. "we were able to consistently analyze the CP map".

Linguistic issue: can be scientific analysis not consistent?

We removed the word consistently as requested. We rephrased clearer:

Using our recently developed statistical framework for the annotation of molecular map features[22], i.e. confidence maps that assist in the assignment of atomic models within weaker cryo-EM map density, we were able to analyze the CP map from Ca²⁺/acidic pH and water samples in a comparative manner (Figure 2a).

3. "the lower radius region differs significantly..." Please provide in the text the level of significance.

The significance level was 1% FDR. We rephrased the sentence and incorporated the significance level:

Although the two determined TMV maps are very similar for most of the CP, the lower radius region differs significantly at an FDR of 1% (Figure 2a right).

4. "Figure EV2b) revealed that previous studies only poorly resolved this part of the protein as the map values were much weaker than other parts of the protein." It is understandable, that the authors were following the request of the first reviewer, however it did not make the meaning of the sentence more clear. Please rephrase the sentence, it sounds confusing "...as the map values were much weaker than other parts of the protein.." "...the map values..." -> it is not understandable what do you mean if you compare with parts of the protein. It is a linguistic problem -> electron density?

We deleted the respective part of the sentence, as the statement is not necessary for understanding and could be misunderstood. However, use of the suggested term electron density with respect to cryo-EM maps is not correct (see Referee #1 point 1), as the electron scattering of the object atoms results in the Coulomb potential, i.e. the combined contribution of nucleus and electrons, and not, as in X-ray crystallography, the electron density.

5. "... regions revealed that the protein backbone follows an alternative path." Please explain the difference between paths, what does it mean "alternative", what was the difference.

Perhaps the word alternative was confusing. We changed “alternative” to a “different” path. In this part of the Results section, we state that there is a difference in the cryo-EM density:

Detailed comparison of the Ca²⁺/acidic pH and water structure at the lower radius region showed that the protein backbone follows a different path (Figure 2b top).

In the following sentences on page 6, we did describe the actual differences in an entire paragraph of the manuscript in more detail with respect to the modelled atomic models.

6. "The multiple models placed for the water structure in the lower radius region deviate to a smaller but significant extent around 1.0 Å"

Was it only in specific regions? Possibly will be useful to indicate them by numbering aa?

For reference, we did describe the region of difference in the sentence before in the manuscript:

“In the water structure at lower radius, however, we identified 3 co-existing models in the residue range 97 – 100 that describe the map, e.g. the map of the water structure is consistent with multiple conformations of E97 (Figure 2b right).”

For clarity, we add to the revised version the E97 – A100 residue range in parenthesis at the commented sentence.

The multiple models placed for the water structure in the lower radius region (E97 – A100) deviate to a smaller but significant extent around 1.0 Å (Table 2).

7. "Annotation of the Ca²⁺ ion is based on the strong map intensities at the respective site and octahedral coordination"

Possibly it did not help to follow recommendations of the first reviewer "map intensities" -> electron densities?

Please see point 4 for discussion of the issue of using the term electron densities in the context of cryo-EM. Perhaps the confusion arose from the term intensities as it is used in X-ray crystallography. Therefore, we changed “strong map intensities” to “high map values”.

8. "... be noted that due to the convoluted nature of conformational changes at the lower radius regions" What is it "...the convolution nature...of changes". Please have respect for an average reader.

We replaced “convoluted” with “comprehensive” as requested. For reference, the comprehensive nature was already described in the sentence before:

The involvement of additional residues such as N101 and N98 in the coordination of the Ca²⁺ binding site suggests a more intricate conformational network responsible for rearrangements beyond the previously postulated carboxylate pair repulsion driving disassembly,..”

In the revised version it reads:

It should be noted that due to the comprehensive nature of conformational changes at the lower radius regions, it is not possible to assign a temporal order to the conformational changes from the two observed structural states of TMV assembly.

9. "For each local window we used a hyperbolic tangent low pass filter with a fall-off of 0.1 and cutoff frequency given by the local resolution." It is an interesting filter, could you provide a profile with parameters on it? Again, please consider an average reader.

It is not common in EM publications to provide this detail of detail as the described procedure refers to standard image filtering operations.

The filter function is given as:

$$filter(f) = 0.5 \left(\tanh \left(\frac{\pi(f + f_H)}{2af_H} \right) + \tanh \left(\frac{\pi(f - f_H)}{2af_H} \right) \right)$$

where a is the falloff (for us $a=0.1$), f the frequency and f_H the filter frequency. The plot below shows the filter function as a function of the frequency in absolute frequency units, which is a frequency measure independent of the pixel size and can be calculated from the spatial frequency by $f_{absolute} = f_{spatial} * apix$. E.g. the Nyquist frequency, the highest frequency than can be resolved, has an absolute frequency unit of 0.5. The plotted filter function has a falloff of 0.1 and a filter frequency of 0.4.

10. "Mg²⁺ was placed in proximity of the RNA and justified by the known tendency of RNA to be stabilized by Mg²⁺ ions. The Ca²⁺ ion was identified by the combination of high map intensities, octahedral coordination and lower B factors in the respective region (Figure EV5), which distinguishes it clearly from water molecules. The Ca²⁺ ion was only found in the cryo-EM structure with high Ca²⁺ concentrations."

This is a very useful piece of information. How much was the difference in the local map densities corresponding to water and Mg or Ca ions. How was it assessed?

Based on visual analysis and adjusting the threshold of the Ca²⁺/acidic pH map, the peak that we assigned to Ca²⁺ has map values of up to 0.07, while Mg²⁺ up to 0.045 and most waters only up to 0.03. For TMV in water, we see the Mg²⁺ peak with values of approximately 0.06, while most waters again have peak values at around 0.03. These peak differences are expected from the differences in electron scattering cross sections of the respective atoms.

It is not common to provide this level of detail in EM publications as the precise amplitudes that result from cryo-EM experiments are not as accurately determined as in X-ray crystallographic experiments.

It is not common to provide this level of detail in EM publications as the precise amplitudes that result from cryo-EM experiments are not as accurately determined as in X-ray crystallographic experiments.

Acceptance

22 August 2019

I am very pleased to accept your manuscript for publication in the next available issue of EMBO reports. Thank you for your contribution to our journal.

Corresponding Author Name: Carsten Sachse

Manuscript Number: EMBOR-2019-48451